# Cross-Sectional and Longitudinal Associations between Non-School Time Physical Activity, Sedentary Time, and Adiposity among Boys and Girls: An Isotemporal Substitution Approach

**DOI:** 10.3390/ijerph18094671

**Published:** 2021-04-27

**Authors:** Kelsey L. McAlister, Jennifer Zink, Daniel Chu, Britni R. Belcher, Genevieve F. Dunton

**Affiliations:** 1Department of Preventive Medicine, University of Southern California, Los Angeles, CA 90032, USA; kmcalist@usc.edu (K.L.M.); jennifaz@usc.edu (J.Z.); chudanie@usc.edu (D.C.); dunton@usc.edu (G.F.D.); 2Department of Psychology, University of Southern California, Los Angeles, CA 90032, USA

**Keywords:** reallocation, obesity, youth, sex differences, accelerometry

## Abstract

This study investigated the cross-sectional and longitudinal associations of the substitution of non-school time light physical activity (LPA), moderate-to-vigorous physical activity (MVPA), and sedentary time (ST) with adiposity in boys and girls. Boys (*n* = 65, baseline *M*age= 9.93 ± 0.86 years) and girls (*n* = 77, baseline *M*age = 10.17 ± 0.95 years) wore waist-worn accelerometers (ActiGraph GT3X) at baseline and at a 30-month follow-up, from which non-school time LPA, MVPA, ST, and total device wear were quantified. Body mass index (BMI) and waist-to-height-ratio (WHR) were measured at baseline and follow-up. Body fat percent (BF%) was obtained at follow-up only. Isotemporal substitution models assessed the cross-sectional and longitudinal associations of reallocating non-school time activity with BMI, WHR and BF%. In boys, replacing 30 min/day of LPA with MVPA was cross-sectionally (β = −8.26, *p* < 0.05) associated with a lower BF%. Replacing 30 min/day of ST with MVPA was cross-sectionally (β = −6.02, *p* < 0.05) associated with a lower BF% in boys. Longitudinally in boys, replacing 30 min of change in LPA with MVPA (β = −7.42, *p* < 0.10) and replacing 30 min of change in MVPA with ST (β = 5.78, *p* < 0.10) over 30 months was marginally associated with less BF%. Associations were null in girls (*p* > 0.05). These results may support targeting activity reallocation during non-school time for the purposes of adiposity improvement in boys. A multi-behavioral approach may be more appropriate for girls, as non-school time activity may not be driving adiposity status.

## 1. Introduction

One of the highest prevalence rates of obesity in the United States is among youth aged 6–11 years [1]. Youth with overweight or obesity have a 6-fold greater risk of having overweight or obesity as an adult, compared to their healthy weight counterparts [2], a finding that is supported by several tracking studies [3,4,5,6]. Unhealthy adiposity during childhood, such as excess weight and high total body fat, predisposes youth to adverse health outcomes as adults, including dyslipidemia, insulin resistance, hypertension, metabolic syndrome and inflammation [2,7]. Furthermore, studies among youth suggest that excess body fat around the waist, specifically, is associated with poor health outcomes and can uniquely contribute to hypertension, hyperlipidemia, insulin resistance, and overall cardiometabolic risk [8,9]. 

Due to the well-documented health consequences associated with obesity during childhood, understanding the contributions of modifiable behaviors, such as physical activity (PA) and sedentary time (ST), to adiposity has been of interest. Moderate-to-vigorous PA (MVPA) is consistently and inversely associated with adiposity markers such as waist circumference, body mass index (BMI), and total body fat in youth [10,11]. Evidence also demonstrates that light PA (LPA) is inversely associated with body fat, skinfold thickness, and BMI in youth [12]. However, the associations between ST and various adiposity markers have been inconsistent [11,13]. Many of these associations between ST and adiposity are attenuated when adjusting for MVPA [13,14], making it unclear how the combination of behaviors influence adiposity. 

Since the amount of time within the day is finite, time spent on one activity displaces time spent in another. Isotemporal substitution is a statistical approach that mathematically estimates the associations of reallocating one type of activity with another of equal time [15]. Results of isotemporal substitution models yield estimated associations of activity reallocation, which may be easier for practitioners to understand and implement. Studies utilizing isotemporal substitution have found associations between the reallocation of activity and various adiposity markers in youth when analyzing activity across the entire day [16,17,18,19]. However, evidence suggests that youth participate in low levels of PA during school [20,21,22], possibly because youth do not necessarily have volitional control over activity during structured school-time. Additional evidence supports that youth do not compensate for lack of school-time PA outside of school [23,24], indicating that this may be a crucial time of day to target interventions. The effects of reallocating time spent sedentary to LPA and MVPA on adiposity has yet to be studied during time outside of school, which is when youth may have less structure and more autonomy over their health behaviors. An additional limitation of many previous studies is that they failed to account for the differences in body fat accumulation and distribution between boys and girls [25,26], thus limiting our understanding of whether or not findings are generalizable to both sexes or are sex-specific. Understanding sex-specific associations could provide insight into tailored intervention strategies aimed at increasing PA and decreasing adiposity, particularly during the adolescent transition, which has sex-specific developmental milestones [27]. Given the developmental differences observed in boys and girls around this age [25,26], we performed exploratory analyses with the purpose of describing cross-sectional and longitudinal associations of substituting non-school time (i.e., time outside of school-time) ST, LPA, and MVPA on adiposity in boys and girls. 

## 2. Materials and Methods

### 2.1. Study Design, Recruitment, and Participants

Data were obtained from the Mother’s and Their Children’s Health (MATCH) Study, a longitudinal study examining stress, parenting factors, health behaviors, and obesity in mother and child dyads. A convenience sample was obtained between August 2014 and March 2016. Children ages 8–12 years were recruited from public elementary schools and after-school programs in the greater Los Angeles area. Children were recruited via informational flyers and in-person recruitment events. Families were included in the study if (1) the child was in third–sixth grade at study enrollment; (2) ≥50% of the child’s custody was with the mother; and (3) the child was able to read English. Families were excluded from the study if (1) the child was currently taking medications for thyroid function or psychological conditions; (2) the child had health issues that limited PA participation; (3) the child was enrolled in special education programs; (4) the mother was pregnant; (5) the child was using oral or inhalant corticosteroids for asthma; (6) the child was classified as underweight by a BMI percentile <5% (adjusted for sex and age); and/or (7) the mother was working more than two weekday evenings (between the hours of 5–9 p.m.) per week or more than 8 h on any weekend day. 

### 2.2. Procedures

A detailed description of study procedures is reported elsewhere [28]. Briefly, parental consent and written child assent were obtained from each participant. Six waves of data collection were conducted at approximately 6-month intervals. Analyses for the present study included data from Wave 1 (baseline) and Wave 6 (+30 months). Children attended a 90 min, in-person data collection visit at each wave, either at a local school or community center. During the in-person visits, the participants completed paper-and-pencil questionnaires, underwent anthropometric measurements, and were given a waist-worn accelerometer to wear for seven consecutive days. Total body fat percent (BF%) was obtained only during the last wave (30 months). Children were compensated USD 100 for completing procedures at each wave. The Institutional Review Board at the University of Southern California approved of the study. 

### 2.3. Measures

#### 2.3.1. Demographic and Participant Characteristics

Each mother completed questionnaires to report their highest level of education achieved (college degree or higher: yes vs. no) and their child’s race/ethnicity (Hispanic: yes vs. no); children self-reported their date of birth and sex. Age in years was calculated based on the time between date of birth and the date of their baseline and 30-month visit. These covariates were selected due to their evidence of confounding activity–adiposity associations [29,30].

#### 2.3.2. Adiposity Markers

Height was measured to the nearest 0.1 centimeter (cm) and weight to the nearest 0.1 kilogram (kg) using a professional stadiometer (Seca 213 Portable Stadiometer) and an electronically calibrated digital scale (Tanita WB-110A), respectively. Height and weight were measured in duplicate, with the average used in analyses. From these measurements, BMI (kg/m^2^) at baseline and 30 months were calculated. Waist circumference was measured using a standard tape measure and following the National Health and Nutrition Examination Survey (NHANES) protocol [31]. Waist circumference was measured at the superior iliac crest at the end of expiration to the nearest 0.1 cm and was measured in duplicate. A third measure was taken if the measures did not fall within 1 cm. The average of all waist circumference measurements was used in the analyses [31]. Waist-to-height ratio (WHR) was computed for each child at baseline and at 30 months. At the 30-month follow-up, a bioelectrical impedance analysis scale (Tanita WB-110A) was used to obtain BF% in each child participant, which has been previously validated for use in youth [32,33]. 

#### 2.3.3. Objective Non-School Time Physical Activity and Sedentary Time

Participants were instructed to wear a waist-worn triaxial ActiGraph GT3X accelerometer (Penscacola, FL, USA) during all waking hours except for bathing and swimming for seven consecutive days. Data were collected in 30 s epochs. Similar to national studies in youth, periods of non-wear were considered >60 continuous minutes of zero counts [34,35,36]. School hours (8:00 a.m. to 3:00 p.m. during Monday–Friday) were removed from the accelerometer data prior to the calculation of all day-level summaries. Thus, non-school time activity was considered as any activity that occurred prior to 8:00 a.m. and after 3:00 p.m. on all weekdays and any activity on weekend days. As BF% was only collected at 30 months, all cross-sectional analyses were conducted with follow-up data only. For cross-sectional analyses, non-school time wear compliance was defined as ≥4 h/day of wear during designated non-school time (i.e., hours before 8:00 a.m. and after 3:00 p.m.) on weekdays and ≥10 h/day of wear on weekend days for one or more days at the follow-up. For longitudinal analyses, valid non-school time wear compliance included wearing the accelerometer for one or more days at both the baseline and follow-up. Although the use of one or more valid days may be considered a liberal protocol, it aided in maximizing the sample size and ensured the inclusion of youth of various adiposity statuses, since previous evidence in youth suggests that a stricter valid number of days criteria excludes youth with overweight or obesity [37]. Another study in youth has also used this protocol [38], and a majority of youth included in the analytic sample had four or more valid accelerometer days (cross-sectional analyses: *n* = 109 at follow-up; longitudinal analyses: *n* = 121 at baseline and *n* = 103 at follow-up). Cut-points for time spent in LPA, MVPA, and ST were derived using age-specific thresholds for youth (adjusted for 30 s epochs) using the Freedson prediction equation equivalent to 4 metabolic equivalents (METs), which were consistent with national studies [34,39]. ST was defined as <100 activity counts per minute [40]. Non-school time mean LPA, MVPA, ST, and accelerometer wear-time in minutes per day were calculated for each participant and used in the analyses. 

### 2.4. Statistical Analysis

All analyses were performed in SAS version 9.4 (SAS Institute Inc., Cary, NC, USA). Due to sex differences in the development of adipose mass, accumulation, and distribution [27], analyses were stratified by sex. Descriptive statistics, including the mean and standard deviation for continuous variables and frequency and the percent of categorical variables, were calculated for the analytic sample. Independent sample t-tests and chi-square tests were used to determine significant differences in continuous and categorical demographic, adiposity, and activity characteristics between those included and excluded in the analyses, respectively. Paired sample t-tests and independent samples t-tests were used to test the differences in continuous demographic, adiposity, and activity characteristics between baseline and 30-month follow-up and between boys and girls, respectively. Chi-square analyses were used to test the differences between categorical participant demographic variables between boys and girls. 

Linear regression models using isotemporal substitution [15] were used to identify the cross-sectional and longitudinal associations of replacing one type of non-school time activity (LPA, MVPA, ST) with another on BMI, WHR, and BF% in boys and girls. Since sleep data were not collected, the resulting estimates were constrained to waking non-school time (waking time before 8:00 a.m. and after 3:00 p.m. on weekdays and any waking time on weekends). Mean non-school time LPA, MVPA, ST, and total accelerometer wear-time (LPA + MVPA + ST = total wear-time) in minutes per day were divided by a constant of 30, similar to other studies using isotemporal substitution analyses [19,41]. Dividing by 30 allowed for estimates to be interpreted so that a one unit increase in each non-school time activity was representative of an increase of 30 min/day. Additionally, we ran ancillary analyses using 15 min substitutions (see Appendix A). Due to the structure of the isotemporal substitution models, the *p*-values for the 15 min reallocation models are equivalent to the 30 min models; however, the beta estimates represent the associations for a smaller amount of activity replacement. In each isotemporal substitution model, the activity of interest (the one being replaced) was removed from the model (e.g., ST), with all other activities (e.g., LPA and MVPA), total wear-time, and other covariates remaining. By removing the activity of interest, the total wear-time then represents the estimate of the activity removed while also acting as a covariate. The corresponding coefficients represent the association of replacing the activity removed with the other activity, while keeping all other variables constant [15]. Cross-sectional models were adjusted for total wear-time and the a priori covariates mentioned above [15,29,30]. 

Longitudinal isotemporal substitution models were fit such that estimates reflected the association of replacing the change from baseline to 30 months in non-school time activity on BMI, WHR, and BF% at the 30-month follow-up. The mean change in minutes per day for non-school time LPA, MVPA, and ST from baseline to 30 months (i.e., activity at follow-up−activity at baseline = change in activity) was calculated. The mean change for non-school time LPA, MVPA, and ST were each divided by a constant of 30 to capture the associations of replacing a 30 min change from baseline to follow-up of one activity with a 30 min change of another. For example, the models that removed the ST estimated the associations between replacing a 30 min change of ST over 30 months with a 30 min change in LPA and MVPA and each adiposity measure at the 30-month follow-up. Similar to cross-sectional models, additional longitudinal isotemporal substitution models estimating the associations between 15 min activity substitutions and adiposity were conducted (see Appendix A). Longitudinal models were adjusted for a priori covariates [29,30] and additionally adjusted for activity levels and accelerometer wear-time at baseline. In order to satisfy the model assumption of homoscedasticity, log BMI was used in cross-sectional and longitudinal models. Linearity and homoscedasticity assumptions were met in cross-sectional and longitudinal models with WHR and BF%. Bivariate Pearson correlations were used to check for collinearity between the activity variables used in the models. Significance was set at *p* < 0.05, and marginal significance was set at *p* < 0.10. Cohen’s f^2^ effect size was calculated. G*Power version 3.1 was used to conduct post hoc sample size estimations and power analyses [42].

## 3. Results

### 3.1. Data Availability and Characteristics of Analytic Sample

A total of 202 children were enrolled in the MATCH Study. For cross-sectional analyses, 152 children completed the study procedures at 30 months, of which 143 provided at least one valid day of accelerometer wear. Of the 143 participants, one was missing ethnicity and mother’s highest level of education, and 12 were missing BF%. These exclusions yielded an analytic sample of 142 for models with BMI and WHR and 130 for models with BF% for cross-sectional analyses. Longitudinally, 134 children had at least one valid day of accelerometer wear at baseline and 30 months and were included in models. Independent samples t-tests and chi-square analyses showed no differences in child age, sex, ethnicity, mother’s highest level of education, BMI, and WHR at baseline between those included vs. excluded from the analyses (*n* = 130–142 participants included vs. *n* = 60–72 participants excluded in cross-sectional models, *n* = 122–134 participants included vs. *n* = 68–80 participants excluded in the longitudinal models; all *p*’s > 0.10).

Table 1 presents demographic and non-school time activity characteristics in the analytic sample at baseline, 30 months, and the change across 30 months by sex. Of the 142 children, 45.78% (65/142) were boys and 54.23% (77/142) were girls. The mean (SD) BMI for boys was 18.98 kg/m^2^ (4.01) at baseline and 20.69 kg/m^2^ (4.86) 30 months later. Similarly, girls had a mean (SD) BMI of 19.00 kg/m^2^ (4.01) at baseline and 21.44 kg/m^2^ (4.85) at follow-up. The mean (SD) BF% for boys was 16.86% (10.04), which was significantly lower (*p* < 0.01) than the mean (SD) BF% girls, which was 24.14% (10.10). Boys also accumulated more mean daily minutes of MVPA compared to girls at baseline and follow-up (both *p*’s < 0.01). Boys and girls both had significant increases in BMI and decreases in LPA and MVPA from baseline to 30 months (all *p*’s < 0.01). Girls, but not boys, additionally had a significant increase in ST across the 30 months (*p* < 0.01).

### 3.2. Cross-Sectional Associations of Replacing Time in LPA, MVPA, and ST with Adiposity

Table 2 and Table 3 show cross-sectional associations of replacing 30 min of each non-school time activity for one another on BMI, WHR, and BF% in boys and girls. In boys, replacing 30 min/day of LPA with 30 min of MVPA was marginally associated with lower BMI (β = −0.13, 95% CI −0.27, 0.01, *p* ≤ 0.10), lower WHR (β = −0.04, 95% CI −0.08, 0.01, *p* ≤ 0.10), and significantly associated with less BF% (β = −8.26, 95% CI −15.42, −1.09, *p* < 0.05). In boys, replacing 30 min/day of MVPA with 30 min of ST was significantly associated with more BF% (β = 6.02, 95% CI 0.49, 11.55, *p* < 0.05). Given the structure of the isotemporal substitution, the estimates were inversed when replacing MVPA with LPA, ST with LPA, and ST with MVPA with the same *p*-value (see Table 2 and Table 3). All other associations in boys were null (all *p*’s > 0.10). In girls, replacing 30 min/day of LPA with MVPA or ST was not cross-sectionally related to BMI, WHR, or BF% (all *p*’s > 0.10). Post hoc sample size calculations showed that 62 participants were needed to achieve a power of 0.8 and a medium effect size (Cohen’s f^2^) of 0.25. The cross-sectional isotemporal substitution models yielded a Cohen’s f^2^, an indicator of effect size, of 0.20–0.40 and post hoc power analyses indicated that models yielded a power of 0.85–0.99.

### 3.3. Longitudinal Associations Replacing Time in LPA, MVPA and ST on Adiposity

Table 4 and Table 5 show longitudinal estimates for replacing a 30 min change of LPA, MVPA, and ST over 30 months on BMI, WHR, and BF% at the 30-month follow-up. In boys, replacing a change of 30 min of LPA with a change of 30 min of MVPA over 30 months was marginally associated with less BF% (β = −7.42, 95% CI −15.44, 0.60, *p* ≤ 0.10). Replacing a change of 30 min of MVPA with a change of 30 min of ST in boys was marginally associated with more BF% (β = 5.78, 95% CI −0.36, 11.93, *p* ≤ 0.10). Similar to the cross-sectional models, estimates were inversed when replacing MVPA with LPA and ST with MVPA (see Table 4 and Table 5). All other longitudinal associations in boys were null (all *p*’s > 0.10). All longitudinal associations in girls were null (all *p*’s > 0.10). Post hoc sample size calculations indicated that 72 participants were needed to achieve a power of 0.8 and a medium effect size (Cohen’s f^2^) of 0.25. The longitudinal isotemporal substitution models yielded a Cohen’s f^2^ of 0.33–0.54 and a power of 0.81–0.98.

## 4. Discussion

This study used isotemporal substitution to identify whether replacing non-school time LPA, MVPA, and ST with equivalent time in each corresponding activity was cross-sectionally and longitudinally associated with adiposity separately in boys and girls. Similar to other studies [44,45], boys and girls had significant decreases in non-school time LPA and MVPA from baseline to 30 months, and girls additionally had an increase in non-school time ST. Our main findings suggest that replacing 30 min/day in LPA with MVPA and replacing 30 min/day of ST with MVPA were cross-sectionally associated with decreases in BF% in boys; however, these associations were marginal in longitudinal models. Favorable but marginal associations were observed for replacing 30 min/day of LPA with MVPA on BMI and WHR outcomes in boys cross-sectionally, but not longitudinally. Contrary to expected directions, replacing 30 min/day of LPA with ST was marginally related to lower BMI, WHR, and BF% in boys in cross-sectional analyses. All cross-sectional and longitudinal replacement estimates were nonsignificant among girls. 

To our knowledge, this is one of the first studies to find favorable cross-sectional associations between activity reallocation and body fatness in boys outside of school-time and during puberty. Longitudinal results suggested that 30 min reallocations in change in activity over a 30-month period may benefit body fatness in boys at 30 months. Replacing 30 min/day of non-school time LPA or ST with MVPA could counteract age-related reductions in MVPA [44,46] and could benefit BF% in boys. Only one other study appears to have used isotemporal substitution among boys and girls separately, finding that replacing entire-day LPA or ST with MVPA was associated with lower fat mass indices in both sexes ages 9–11 years [41]. Similarly, Collings et al. found that substituting ST with light, moderate, or vigorous PA was inversely associated with fat and trunk fat indices in younger youth [47]. Other studies of activity levels across the entire day in youth aged 6–11 years suggested that reallocating ST to LPA and MVPA could improve multiple measures of body fatness [17,18]. However, while our findings are consistent with some prior studies, our models were marginally statistically significant.

An important distinction between our analyses and prior studies is that we only included non-school time activity. Youth may be more limited in their ability to modify movement behaviors during designated school-time, presenting challenges for activity reallocation recommendations. Although school-based interventions have been successful in increasing school-time PA participation [48], evidence suggests they generally do not increase PA outside of school-time; which may be one reason why school-based activity interventions yield minimal effects on adiposity [48,49]. Additionally, youth may become more independent in their health decisions as they age into adolescence [50], and evidence shows that youth spend a majority of after-school time being sedentary [51]. Establishing healthy habits by finding strategies to increase MVPA outside of school may be a method for reducing body fatness. Our results indicate that interventions that implement 30 min reallocations of ST to LPA or MVPA outside of school-time may potentially be effective in reducing overall body fat in boys. 

Our cross-sectional findings suggest that replacing 30 min/day of LPA with MVPA may relate to lower BMI in boys. This association may be transient because it was not observed longitudinally; the reallocation of activity may influence BMI proximally, rather than over this 30-month period of time. Other studies utilizing isotemporal substitution modeling found cross-sectional and prospective associations between activity replacement and BMI in youth of similar age [52]. Although BMI is frequently used in the literature due to its ease of measurement [53], it may not be the best measure of adiposity, particularly among pubertal youth, as it fails to capture adiposity beyond generalized body proportions and does not capture body fat distribution [54]. Considering that BMI provides a limited scope, our findings regarding BF% in boys provide stronger health implications for activity reallocation. 

High central adiposity is associated with adverse health outcomes in youth, including increased cardiometabolic risk [9]. We found that among boys, replacing 30 min/day of LPA with MVPA may relate to lower WHR. Although our finding was marginal, the association of replacing LPA with MVPA on WHR is consistent with the strong evidence that supports a dose-response relationship between PA intensity and abdominal obesity [55]. It has been suggested that higher intensity PA is more effective in triggering the secretion of lipolytic hormones, promoting greater post-PA energy expenditure and fat oxidation, and supporting greater negative energy balance—all aspects that contribute to lower visceral abdominal fat [56]. Future studies should consider analyzing the relationship between reallocating time spent in LPA with MVPA and central adiposity since our study was only able to find a marginally significant association.

Unexpectedly, the isotemporal substitution models suggested that replacing LPA with ST would have favorable effects on BMI, WHR, and BF% in boys cross-sectionally. These findings were marginal and could be an artifact of the statistical analyses that were performed. Opposingly, other studies using isotemporal models found that replacing ST with LPA was related with improved adiposity, including lower waist circumference, WHR, fat mass, and BMI [16,18,52]. Our marginally significant findings counteract the traditional thought that any type of PA is better than ST, and these contradictory findings may be attributed to the possibility that only certain types of ST are related to adiposity [57,58]. Previous studies found that sedentary behaviors such as screen-time and television watching were associated with higher adiposity, whereas homework and reading were not [57,58]. The type of ST, which is not captured by accelerometers, may be an important explanatory factor, therefore future studies should consider gathering the type of ST using validated self-report measures to distinguish possible contextual differences. 

Interestingly, we only found associations between activity reallocation and adiposity outcomes in boys. This could be because adiposity changes among girls in this age group may not be entirely attributable to changes in PA and ST, rather they could be due to changes in growth, development, and puberty-related hormones [59]. Anthropometric studies indicate that sex-specific patterning of fat emerges during puberty, suggesting that girls tend to carry less fat around the waist but more around the hips compared to boys [60,61]. Studies using direct measures of body fat demonstrated that girls have more fat than boys prior to the onset of puberty [25,26]. Higher body fat prior to puberty combined with decreases in PA and increases in ST in girls of this age may indicate that girls need larger amounts of time in PA than in ST compared to boys. Therefore, girls in this age group during this time may require more than 30 min/day in behavioral changes, or even a multifaceted approach that targets multiple activity behaviors and diet, to elicit favorable changes in body fat.

A strength of our study is that it captures youth during a unique developmental period in their life. During the 30 months of follow-up for this age, youth were undergoing the transition through puberty and had decreases in PA and increases in ST [38]. Non-school time PA and ST were measured via accelerometry, which provided objective measures of body movement. We used isotemporal substitution models to provide mathematical estimations of reallocation associations of ST, LPA, and MVPA outside of school-time on adiposity. We focused on non-school time activity because it is understudied and more likely to be under the youth’s control. Furthermore, this study contributes to the literature by showing that activity reallocation can potentially benefit adiposity in boys.

This study also has limitations. Although our models yielded sufficient statistical power, thus it is possible that some of the models were underpowered given the small sample size. Sample size calculations suggested we needed 62–72 participants and some of our analyses only had 57–65 participants. This also may have contributed to our higher number of marginally significant compared to significant findings. A known limitation of the isotemporal substitution models is multicollinearity given the number of activity variables that are entered into the models [15]. The highest correlation in our study was between ST and total wear-time (*r* = 0.89), which was expected considering that ST made up most of the total wear-time. However, there was relatively little intercorrelation between the other activity variables (*r* ranging from 0.01–0.74). Although we collected self-reported puberty data, we opted not to control for puberty due to the large amount of missing data (*n* = 38 missing at baseline; *n* = 36 missing at follow-up). Puberty is important for fat accumulation [62] and therefore should be measured and included in isotemporal substitution models in future studies. There were additional covariates (e.g., dietary intake, sleep) not controlled for in our analyses known to influence adiposity that future studies may want to explore [63]. We were unable to establish causality due to the observational nature of this study. Furthermore, accelerometry does not capture activities such as biking, upper-arm movements, and swimming, so it is possible there were activities unaccounted for [64]. Lastly, we were unable to distinguish the mode of sedentary and physical activities participated in; future studies should consider gathering the context and type of activity to provide more insight for interventions. 

## 5. Conclusions

This study found that reallocating 30 min/day of non-school time LPA to MVPA and ST to MVPA was cross-sectionally related to lower BF% in boys; longitudinal associations were marginal. We additionally found marginal cross-sectional associations between replacing 30 min/day of non-school time LPA with MVPA and WHR and BMI in boys. However, our study was unable to demonstrate reallocation associations in girls, suggesting that fat accumulation in girls may be a result of a combination of other factors that are not yet understood. As our study focused on non-school time activity behaviors during the critical pubertal transition, it offers insight into sex-specific targeted intervention strategies. Future interventions in boys should consider targeting non-school time activity displacement for the purposes of adiposity control, which could therefore counteract the adverse consequences of high childhood adiposity. Future studies in girls should consider using a multi-behavioral approach, as it appears that activity outside of school-time may not be the driving factor in adiposity status.

## Figures and Tables

**Table 1 ijerph-18-04671-t001:** Participant characteristics for the analytic sample (*n* = 142; boys *n* = 65; girls *n* = 77).

	Boys			Girls		
	Baseline ^a^	30-Month Follow-Up	Change from Baseline to Follow-Up ^a^	Baseline ^d^	30-Month Follow-Up	Change from Baseline to Follow-Up ^e^
	Mean ± SD	Mean ± SD	Mean ± SD	Mean ± SD	Mean ± SD	Mean ± SD
Age (years)	9.93 ± 0.86	12.40 ± 0.91	2.43 ± 0.19 **	10.17 ± 0.95	12.57 ± 0.94	2.38 ± 0.19 **
BMI (kg/m^2^)	18.98 ± 4.01	20.69 ± 4.86	1.85 ± 1.82 **	19.00 ± 4.01	21.44 ± 4.85	2.24 ± 1.84 **
WHR	0.49 ± 0.06	0.49 ± 0.06	−0.001 ± 0.04 ^b^	0.50 ± 0.07	0.51 ± 0.06	0.06 ± 0.05 ^e^
BF% (%)	--	16.86 ± 10.04 ^c^^	--	--	24.14 ± 10.10 ^d^^	--
LPA (min/day)	157.21 ± 34.29	134.06 ± 45.13	−22.04 ± 54.48 **	166.25 ± 38.66	132.81 ± 50.77	−30.72 ± 44.51 **
MVPA (min/day)	36.25 ± 19.67 ^	21.27 ± 16.26 ^	−13.62 ± 19.64 **	27.50 ± 17.40 ^	15.17 ± 11.93 ^	−11.39 ± 18.99 **
ST (min/day)	298.28 ± 72.70	305.39 ± 77.66	6.07 ± 105.45	281.12 ± 67.18	321.36 ± 72.78	34.66 ± 89.00 **
Total wear-time (min/day)	491.74 ± 90.63	460.72 ± 97.29	−29.59 ± 139.82	474.87 ± 77.79	469.34 ± 89.36	−7.45 ± 100.48
	*n* (%)	*n* (%)		*n* (%)	*n* (%)	
Ethnicity						
Hispanic	28 (44.44)	32 (49.23)	--	44 (60.27)	36 (46.75)	--
Non-Hispanic	35 (55.56)	33 (50.77)	--	29 (39.73)	41 (53.25)	--
Mother’s education level						
Less than a college degree	29 (46.03)	30 (40.15)	--	25 (34.25)	30 (38.96)	--
College degree and above	34 (53.97)	35 (53.85)	--	48 (65.75)	47 (61.04)	--

Abbreviations and acronyms—SD: standard deviation; min: minutes; %: percent; BMI: body mass index; WHR: waist-to-height ratio; BF%: body fat percent; LPA: light physical activity; MVPA: moderate-to-vigorous physical activity; ST: sedentary time. ^a^
*n* = 63; ^b^
*n* = 62; ^c^
*n* = 59; ^d^
*n* = 71; ^e^
*n* = 68; ** statistically significant difference between baseline and follow-up at *p* < 0.01. ^ statistically significant difference between mean in boys vs. mean in girls at *p* < 0.01.

**Table 2 ijerph-18-04671-t002:** Cross-sectional replacement effects (β [95% CI]) at the follow-up of substituting 30 min of non-school time LPA, MVPA, and ST on BMI, WHR, and BF% in boys (*n* = 65).

BMI ^a^				
	With 30 min/day of:			
Replacing 30 min/day of:	LPA	MVPA	ST	Total wear-time
LPA	--	**−0.13** **[−0.27, 0.01] ^#^**	**−0.05** **[−0.10, 0.003] ^#^**	**0.04** **[−0.005, 0.08] ^#^**
MVPA	**0.13** **[−0.01, 0.27] ^#^**	--	0.08 [−0.03, 0.19]	−0.10 [−0.21, 0.02]
ST	**0.05** **[−0.003, 0.10] ^#^**	−0.08 [−0.19, 0.03]	--	−0.01 [−0.03, 0.01]
**WHR**				
	With 30 min/day of:			
Replacing 30 min/day of:	LPA	MVPA	ST	Total wear-time
LPA	--	**−0.04** **[−0.08, 0.01] ^#^**	**−0.01** **[−0.03, 0.01] ^#^**	0.01 [−0.01, 0.02]
MVPA	**0.04** **[−0.01, 0.08] ^#^**	--	0.03 [−0.01, 0.06]	−0.03 [−0.06, 0.01]
ST	**0.01** **[−0.01, 0.03] ^#^**	−0.03 [−0.06, 0.01]	--	−0.002 [−0.01, 0.004]
**BF% ^b^**				
	With 30 min/day of:			
Replacing 30 min/day of:	LPA	MVPA	ST	Total wear-time
LPA	--	**−8.26** **[−15.42, −1.09] ***	**−2.24** **[−4.85, 0.38] ^#^**	1.72 [−0.46, 3.89]
MVPA	**8.26** **[1.09, 15.42] ***	--	**6.02** **[0.49, 11.55] ***	**−6.54** **[−12.29, −0.79] ***
ST	**2.24** **[−0.38, 4.85] ^#^**	**−6.02** **[−11.55, 0.49] ***	--	−0.52 [−1.56, 0.52]

**Bold estimates** represent significant or marginally significant associations. Abbreviations—min: minutes; CI: confidence interval; LPA: light physical activity; MVPA: moderate-to-vigorous physical activity; ST: sedentary time; BMI: body mass index; WHR: waist-to-height ratio; BF%: body fat percent. All models were adjusted for child age, ethnicity, mother’s education, and accelerometer wear-time. ^a^ Log BMI was used to satisfy model assumptions. Estimates are presented in the log transformation of BMI. ^b^
*n* = 59; * *p* < 0.05, ^#^
*p* < 0.10.

**Table 3 ijerph-18-04671-t003:** Cross-sectional replacement effects (β [95% CI]) at follow-up of substituting 30 min of non-school time LPA, MVPA, and ST on BMI, WHR, and BF% in girls (*n* = 77).

BMI ^a^				
	With 30 min/day of:			
Replacing 30 min/day of:	LPA	MVPA	ST	Total wear-time
LPA	--	0.02 [−0.13, 0.17]	0.03 [−0.01, 0.07]	0.003 [−0.03, 0.03]
MVPA	−0.02 [−0.17, 0.13]	--	0.01 [−0.12, 0.13]	0.03 [−0.11, 0.16]
ST	−0.03 [−0.07, 0.01]	−0.01 [−0.13, 0.12]	--	**0.03** **[0.01, 0.05] ****
**WHR**				
	With 30 min/day of:			
Replacing 30 min/day of:	LPA	MVPA	ST	Total wear-time
LPA	--	−0.004 [−0.05, 0.04]	0.01 [−0.01, 0.02]	0.002 [−0.01, 0.01]
MVPA	0.004 [−0.04, 0.05]	--	0.01 [−0.03, 0.05]	−0.003 [−0.04, 0.04]
ST	−0.01 [−0.02, 0.01]	−0.01 [−0.05, 0.03]	--	**0.01** **[0.003, 0.02] ****
**BF% ^b^**				
	With 30 min/day of:			
Replacing 30 min/day of:	LPA	MVPA	ST	Total wear-time
LPA	--	−0.33 [−7.74, 7.09]	1.45 [−0.72, 3.62]	−0.014 [−1.60, 1.57]
MVPA	0.33 [−7.09, 7.74]	--	1.78 [−4.44, 7.99]	−0.34 [−7.02, 6.34]
ST	−1.45 [−3.62, 0.72]	−1.78 [−7.99, 4.44]	--	**1.44** **[0.31, 2.57] ***

**Bold estimates** represent significant or marginally significant associations. Abbreviations—min: minutes; CI: confidence interval; LPA: light physical activity; MVPA: moderate-to-vigorous physical activity; ST: sedentary time; BMI: body mass index; WHR: waist-to-height ratio; BF%: body fat percent. All models were adjusted for child age, ethnicity, mother’s education, and accelerometer wear-time. ^a^ Log BMI was used to satisfy model assumptions. Estimates are presented in the log transformation of BMI. ^b^
*n* = 71; ** *p* < 0.01; * *p* < 0.05.

**Table 4 ijerph-18-04671-t004:** Isotemporal substitution associations (β [95% CI]) using 30 min reallocations between 30-month changes in non-school time LPA, MVPA, and ST and BMI, WHR, and BF% at 30-month follow-up in boys (*n* = 63).

BMI ^a^				
	With 30 min/day of:			
Replacing 30 min/day of:	LPA	MVPA	ST	Total wear-time
LPA	--	−0.10 [−0.26, 0.06]	−0.04 [−0.09, 0.01]	0.03 [−0.02, 0.07]
MVPA	0.10 [−0.06, 0.26]	--	0.06 [−0.07, 0.18]	−0.07 [−0.20, 0.06]
ST	0.04 [−0.01, 0.09]	−0.06 [−0.18, 0.07]	--	−0.01 [−0.03, 0.01]
**WHR**				
	With 30 min/day of:			
Replacing 30 min/day of:	LPA	MVPA	ST	Total wear-time
LPA	--	−0.02 [−0.07, 0.02]	−0.006 [−0.02, 0.01]	0.004 [−0.009, 0.02]
MVPA	0.03 [−0.02, 0.07]	--	0.02 [−0.02, 0.06]	−0.02 [−0.06, 0.02]
ST	0.006 [−0.01, 0.02]	−0.02 [−0.06, 0.02]	--	−0.002 [−0.01, 0.004]
**BF% ^b^**				
	With 30 min/day of:			
Replacing 30 min/day of:	LPA	MVPA	ST	Total wear-time
LPA	--	**−7.42** **[−15.44, 0.60] ^#^**	−1.64 [−4.37, 1.10]	1.23 [−1.09, 3.55]
MVPA	**7.42** **[−0.60, 15.44] ^#^**	--	**5.78** **[−0.36, 11.93] ^#^**	**−6.19** **[−12.55, 0.18] ^#^**
ST	1.64 [−1.10, 4.37]	**−5.78** **[−11.93, 0.36] ^#^**	--	−0.40 [−1.40, 0.59]

**Bold estimates** represent significant or marginally significant associations. Abbreviations—min: minutes; CI: confidence interval; LPA: light physical activity; MVPA: moderate-to-vigorous physical activity; ST: sedentary time; BMI: body mass index; WHR: waist-to-height ratio; BF%: body fat percent. All models adjusted for child age, ethnicity, mother’s education, activity, and accelerometer wear-time at baseline. ^a^ Log BMI was used to satisfy model assumptions. Estimates are presented in the log transformation of BMI. ^b^
*n* = 57. ^#^
*p* < 0.10.

**Table 5 ijerph-18-04671-t005:** Isotemporal substitution associations (β [95% CI]) using 30 min reallocations between 30-month changes in non-school time LPA, MVPA, and ST and BMI, WHR, and BF% at 30-month follow-up in girls (*n* = 71).

BMI ^a^				
	With 30 min/day of:			
Replacing 30 min/day of:	LPA	MVPA	ST	Total wear-time
LPA	--	0.03 [−0.13, 0.19]	0.03 [−0.02, 0.08]	−0.002 [−0.04, 0.04]
MVPA	−0.03 [−0.19, 0.13]	--	0.002 [−0.14, 0.14]	0.03 [−0.12, 0.17]
ST	−0.03 [−0.08, 0.02]	−0.002 [−0.14, 0.14]	--	**0.03** **[0.003, 0.05] ***
**WHR**				
	With 30 min/day of:			
Replacing 30 min/day of:	LPA	MVPA	ST	Total wear-time
LPA	--	−0.01 [−0.05, 0.04]	0.006 [−0.01, 0.02]	0.002 [−0.01, 0.01]
MVPA	0.01 [−0.04, 0.04]	--	0.01 [−0.03, 0.05]	−0.003 [−0.05, 0.04]
ST	−0.006 [−0.02, 0.01]	−0.01 [−0.05, 0.03]	--	**0.01** **[0.001, 0.02] ***
**BF% ^b^**				
	With 30 min/day of:			
Replacing 30 min/day of:	LPA	MVPA	ST	Total wear-time
LPA	--	0.42 [−7.30, 8.14]	1.21 [−1.22, 3.64]	−0.06 [−1.87, 1.74]
MVPA	−0.42 [−8.14, 7.30]	--	0.79 [−5.68, 7.25]	0.36 [−6.58, 7.29]
ST	−1.21 [−3.64, 1.22]	−0.79 [−7.25, 5.68]	--	**1.14** **[−0.10, 2.39] ^#^**

**Bold estimates** represent significant or marginally significant associations. Abbreviations—min: minutes; CI: confidence interval; LPA: light physical activity; MVPA: moderate-to-vigorous physical activity; ST: sedentary time; BMI: body mass index; WHR: waist-to-height ratio; BF%: body fat percent. All models adjusted for child age, ethnicity, mother’s education, activity, and accelerometer wear-time at baseline. ^a^ Log BMI was used to satisfy model assumptions. Estimates are presented in the log transformation of BMI. ^b^
*n* = 65. * *p* < 0.05, ^#^
*p* < 0.10 [43].

## Data Availability

The data used to support the findings in this study are available from the corresponding author upon request.

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
