# Peer review of "Cross-Sectional and Longitudinal Associations between Non-School Time Physical Activity, Sedentary Time, and Adiposity among Boys and Girls: An Isotemporal Substitution Approach"

_ijerph, 2021, doi:10.3390/ijerph18094671_

Round 1
Reviewer 1 Report
Overall opinion: Interesting topic, nicely presented.I have only one specific remark:
Despite the fact that no correlation was found between non-school time physical
activity and adiposity in girls under these study conditions,
I suggest supplementing the abstract with what already appears
in the conclusion that multi-behavioral approach is necessary for
girls (we do not want to discourage them from physical activity).
It is important to mention already here for those who just run through the abstract.
Author Response
- Thank you for your suggestion. We have supplemented the abstract conclusion on Lines 28-29 with the statement, “A multi-behavioral approach may be more appropriate for girls, as non-school time physical activity may not be driving adiposity status.”
Reviewer 2 Report
Minor comments-
- Please mention in the conclusion section advantage of these findings over the previous or already done by the other groups.
- How did author measure the body fat, please include in the material methods.
- It seems that association in girls were null. Please mention the rationale, if possible.
Author Response
Please mention in the conclusion section advantage of these findings over the previous or already done by the other groups.
- Strengths of the present study compared to previous studies are described in the discussion section on Lines 487-488 (“An important distinction between our analyses and prior studies is that we only included non-school time activity”) and Lines 548-556(“A strength of our study is that it captures youth during a unique developmental period in their life. During the 30 months of follow-up for this age, youth were undergoing the transition through puberty and had decreases in PA and increases in ST. Non-school time PA and ST were measured via accelerometry, which provided objective measures of body movement. We used isotemporal substitution models to provide mathematical estimations of reallocation associations of ST, LPA, and MVPA outside of school-time on adiposity. We focused on non-school time activity because it is understudied and more likely to be under the youth’s control. Furthermore, this study contributes to the liter-ature by showing that activity reallocation can potentially benefit adiposity in boys.”). We additionally added a sentence to the conclusion section on Lines 584-586 that states, “Because our study focused on non-school time activity behaviors during the critical pubertal transition, it offers insight into sex-specific targeted intervention strategies.”
How did author measure the body fat, please include in the material methods.
- Body fat percentage was measured via a bioelectrical impedance analysis scale. This measure is described in Lines 125-127, which states, “At the 30-month follow-up, a bioelectrical impedance analysis scale (Tanita WB-110A) was used to obtain BF% in each child participant, which has been previously validated for use in youth.”
It seems that association in girls were null. Please mention the rationale, if possible.
- We discuss potential reasons for null associations in girls in Lines 535-547, which states, “Interestingly, we only found associations between activity reallocation and adiposity outcomes in boys. This could be because adiposity changes among girls in this age group may not be entirely attributable to changes in PA and ST, rather they could be due to changes in growth, development, and puberty-related hormones. Anthropometric studies indicate that sex-specific patterning of fat emerges during puberty, suggesting that girls tend to carry less fat around the waist but more around the hips compared to boys. Studies using direct measures of body fat demonstrated that girls have more fat than boys prior to the onset of puberty. Higher body fat prior to puberty combined with decreases in PA and increases in ST in girls this age may indicate that girls need larger amounts of time in PA than in ST compared to boys. Therefore, girls in this age group during this time may require more than 30 minutes/day in behavioral changes, or even a multifaceted approach that targets multiple activity behaviors and diet, to elicit favorable changes in body fat.”
Reviewer 3 Report
An interesting and needed topic was undertaken in the study of cross-sectional and longitudinal associations between non-school time physical activity, sedentary time, and adiposity among boys and girls, taking into account gender differences, which should be noted.
The work has been carefully prepared in terms of scientific and editorial (with the need to correct the wording of the tables in the text). However, a weakness of the study is showing marginal relationships between non-school time physical activity, sedentary time, and BF%, WHR or BMI and only in boys. Associations were null in girls.
The strength of the study is the research workshop and the discussion of the results, including the strengths and weaknesses of the study, however, studies do not show relationships to be guessed at.
The study may be published as valuable in the category of conclusions for other work to be carried out in the future.
Author Response
- Thank you for your comments regarding our manuscript. We appreciate the time you took to review it.
Reviewer 4 Report
This study uses a unique analysis to determine the effects of replace different levels of intensity for physical activity (or sedentary behaviour) on BMI, waist to hip ratio, and %body fat in boys and girls over a 30 month period.
Line 122: Please indicate whether the bioelectric impedance device used for %fat is validated for use in children.
Line 227: “Boys and girls both had significant changes in BMI, LPA, and MVPA from baseline to 30 months (all p’s<0.01).” Please describe the direction of these changes here (i.e. increase or decrease).
Note that for Tables 2 and 3, the footnotes are misplaced on the pdf (i.e. they appear down the side of the figure).
Line 517: The lack of significant findings in girls was attributable to the variation in changes in body fat that occur during puberty. Were any measures of pubertal status or maturation (i.e. age from peak height velocity, Tanner staging) collected in the study?
Author Response
Line 122: Please indicate whether the bioelectric impedance device used for %fat is validated for use in children.
- We have revised this sentence on Lines 125-127 with additional information: “At the 30-month follow-up, a bioelectrical impedance analysis scale (Tanita WB-110A) was used to obtain BF% in each child participant, which has been previously validated for use in youth.” We also have provided two references for the validation in children. (Hosking et. al., 2006; Lee et al. 2017).
Line 227: “Boys and girls both had significant changes in BMI, LPA, and MVPA from baseline to 30 months (all p’s<0.01).” Please describe the direction of these changes here (i.e. increase or decrease).
- We have revised this sentence on Lines 235-237 to state: “Boys and girls both had significant increases in BMI and decreases in LPA and MVPA from baseline to 30 months (all p’s<0.01).”
Note that for Tables 2 and 3, the footnotes are misplaced on the pdf (i.e. they appear down the side of the figure).
- Thank you for the feedback. We have rearranged the document so that the footnotes are no longer misplaced.
Line 517: The lack of significant findings in girls was attributable to the variation in changes in body fat that occur during puberty. Were any measures of pubertal status or maturation (i.e. age from peak height velocity, Tanner staging) collected in the study?
- Our study did measure pubertal status via a self-reported puberty measure. However, due to the large amount of missing data altogether for our puberty measure (N=38 missing at baseline, N=36 missing at follow-up), we did not control for puberty in our models. This is mentioned on Lines 566-570: “Although we collected self-reported puberty data, we opted not to control for puberty due to the large amount of missing data (N=38 missing at baseline; N=36 missing at follow-up). Puberty is important for fat accumulation and therefore should be measured and included in isotemporal substitution models in future studies”.
Reviewer 5 Report
It is a very interesting research that shows how changing behaviours in the performance of daily activities can benefit the levels of body adiposity in adolescents, both transversely and longitudinally, considering the statistically significant findings and those that were marginally significant. I agree with the authors that the time allotted to different activities outside school hours may be essential to modify behaviour structures on the adherence of the practice of physical activity in adolescents.
Considerations.
Do the authors consider that the data collected between 2014 and 2016 may have some obsolescence?
Do authors consider that compensating participants with money can be unethical? (line 102)
The model, make, accuracy of the metric band must be indicated to determine the waist circumference (line 118)
Please clarify this point. Was the total included compared with the total excluded, which are a smaller number of participants? (line 218)
Cohen statistic and effect size ranges should be indicated in the Materials and Methods section, subsection Statistical Analysis (line 266)
It is important to complement that a limitation of the questionnaires through self-reporting can underestimate or overestimate the responses of the participants (line 548)
Separate paragraphs (line 570)
Thank you.
Author Response
Do the authors consider that the data collected between 2014 and 2016 may have some obsolescence?
- While the data were collected between 2014 and 2016, the authors do not consider it to be obsolete for the following reasons:
- 1) independent of pandemic conditions, the school day is still typically ~8 hours, where they have less volitional control of activity, making the analysis of non-school time activity behaviors important;
- 2) a recent study using national data by Knell et al. (2019) found that a majority of adolescents as recent as the 2017 cycle of the Youth Risk Behavior Surveillance Survey do not meet physical activity guidelines. Additionally, it was estimated that obesity affected 14.4 million children and adolescents in the United States in 2017-2018. Considering that lack of physical activity and high adiposity rates remain an issue in youth, our data provided the opportunity to investigate the associations between activity reallocation and adiposity in the hopes to provide more concrete recommendations that are applicable to today;
- 3) the measurements used in the present study are consistent with current adiposity and physical activity behaviors methodologies;
- 4) the estimates of physical activity collected via the same accelerometer methodologies are in line with estimates from national samples (Belcher, et al., 2010), indicating that overall activity levels have not substantively changed over time;
- 5) The rate of overweight/obesity in our sample was approximately 22% at the 30-month follow-up, which is similar to the rate of obesity reported in the US in 2017-2018, which was approximately 20% for 6- to 11-year-olds and 21% for 12- to 19-year-olds.
- References:
Knell, GK, Durand, CP, Kohl, HW III, Wu, IHC, and Gabrial, KP. 2019. Prevalence and likelihood of meeting sleep, physical activity, and screen-time guidelines among US youth. JAMA Pediatrics, 173(4): 387-389.
Fryar CD, Carroll MD, Afful J. Prevalence of overweight, obesity, and severe obesity among children and adolescents aged 2–19 years: United States, 1963–1965 through 2017–2018. NCHS Health E-Stats. 2020.
Belcher, B. R., et al. (2010). "Physical activity in US youth: effect of race/ethnicity, age, gender, and weight status." Med Sci Sports Exerc 42(12): 2211-2221.
Do authors consider that compensating participants with money can be unethical? (line 102)
- Our study was reviewed by the university Institutional Review Board who approved participant compensation. Compensation was commensurate with the time and effort participants committed in this longitudinal study. Monetary compensation was not the primary recruitment tactic. Therefore, we do not think monetary compensation is the primary motivating factor and do not think it could be seen as coercion. Additionally, evidence indicates that a number of IRBs across many institutions allow for monetary compensation in youth (Kimberly et al., 2006). Bagley et al. (2007) also demonstrated that youth >9 years old show an appreciation for the role and value of money, and concluded that compensating youth 9 years of age and older is appropriate.
- References:
Kimberly, MB, Hoehn, S, Feudtner, C, Nelson, RM, and Schreiner, M. Variation in standards of research compensation and child assent practices” A comparison of 69 institutional review board-approved informed permission and assent form from 3 multicenter pediatric clinical trials. Pediatrics, 117(5): 1706-1711.
Bagley, SJ, Reynolds, WW, and Nelson, RM. 2007. Is a “wage-payment” model for research participation appropriate for children? Pediatrics, 119(1): 46-51.
The model, make, accuracy of the metric band must be indicated to determine the waist circumference (line 118)
- Unfortunately, during data collection, we did not record the make or model of the metric band used to obtain waist circumference. The measuring tape used was a standard measuring tape, and all waist circumference procedures followed the guidelines set forth by the National Health and Nutrition Examination Survey (NHANES). To provide further clarification, we have provided more details on the waist circumference measurement methods in the manuscript. Lines 118-124 states, “Waist circumference was measured using a standard tape measure and following the National Health and Nutrition Examination Survey (NHANES) protocol. Waist circumference was measured at the superior iliac crest at the end of expiration to the nearest 0.1 cm and measured in duplicate. A third measure was taken if the measures did not fall within 1 cm. The average of all waist circumference measurements was used in analyses.”
Please clarify this point. Was the total included compared with the total excluded, which are a smaller number of participants? (line 218)
- We have clarified this point by providing the sample size ranges of those included versus those excluded on Lines 214-220 (“A total of 202 children enrolled in the XX Study (blinded for review). For cross-sectional analyses, 152 children completed study procedures at 30 months, of which 143 provided at least one valid day of accelerometer wear. Of the 143 participants, one was missing ethnicity and mother’s highest level of education, and 12 were missing BF%. These exclusions yielded an analytic sample of 142 for models with BMI and WHR and 130 for models with BF% for cross-sectional analyses. Longitudinally, 134 children had at least one valid day of accelerometer wear at baseline and 30 months and were included in models”). This sentence on Lines 220-225 now states, “Independent samples t-tests and chi-square analyses showed no differences in child age, sex, ethnicity, mother’s highest level of education, BMI, and WHR at baseline between those included vs. excluded from analyses (N=130-142 participants included vs. N=60-72 participants excluded in cross-sectional models, N=122-134 participants included vs. N=68-80 participants excluded in longitudinal models; all p’s>0.10).”
Cohen statistic and effect size ranges should be indicated in the Materials and Methods section, subsection Statistical Analysis (line 266)
- On Line 208 of the Statistical Analysis sub-section, we have added a sentence that states, “Cohen’s f2 effect size was calculated.” Since effect size was used to calculate post-hoc power, we opted to include the yielded Cohen’s f2 ranges in the results section. Lines 273-276 states, “The cross-sectional isotemporal substitution models yielded a Cohen’s f2, an indicator of effect size, of 0.20-0.40 and post-hoc power analyses indicated that models yielded a power of 0.85-0.99.” Lines 389-391 states, “The longitudinal isotemporal substitution models yielded a Cohen’s f2 of 0.33-0.54 and that a power of 0.81-0.98.”
It is important to complement that a limitation of the questionnaires through self-reporting can underestimate or overestimate the responses of the participants (line 548)
- Demographic information was self-reported via questionnaires by the mother. Other than the demographic information, no other variables were collected through self-report. The primary variables of interest (physical activity and measures of adiposity status) were collected through objective measures. Therefore, we do not think a discussion of the limitation of questionnaires is warranted for the present study.
Separate paragraphs (line 570)
- Thank you for pointing out this formatting error. We have separated the paragraphs.